# Training Software Engineering Agents and Verifiers with SWE-Gym

**Jiayi Pan** [* 1]   **Xingyao Wang** [* 2]   **Graham Neubig** [3]   **Navdeep Jaitly** [4]   **Heng Ji** [2]   **Alane Suhr** [† 1]   **Yizhe Zhang** [† 4]

## Abstract

We present SWE-Gym, the first environment for training software engineering (SWE) agents. SWE-Gym contains 2,438 real-world task instances, each comprising a Python codebase with an executable runtime environment, unit tests, and a task specified in natural language. We use SWE-Gym to train language model based SWE agents, and achieve up to 19% absolute gains in resolution rate on the popular SWE-Bench Verified and Lite test sets. We also experiment with inference-time scaling through verifiers trained on agent trajectories sampled from SWE-Gym. When combined with our fine-tuned SWE agents, we achieve 32.0% and 26.0% on SWE-Bench Verified and Lite, respectively, reflecting a new state-of-the-art for open-weight SWE agents. To facilitate further research, we publicly release SWE-Gym, models, and agent trajectories.

## 1. Introduction

Language models (LMs) have remarkable promise in automating software engineering (SWE) tasks, as most clearly measured by recent progress on benchmarks like SWE-Bench (Jimenez et al., 2024) and Commit0 (Zhao et al., 2024). While LM-based SWE agents have shown significant performance gains through improving agent-computer interfaces (Yang et al., 2024) and prompting strategies (Wang et al., 2024c), advances in SWE agents have been limited by a reliance on proprietary models, with limited research to improve the underlying LM itself.

Unlike other domains where supervised fine-tuning and reinforcement learning have significantly improved LM capabilities, such as chat (Ouyang et al., 2022), math reason-

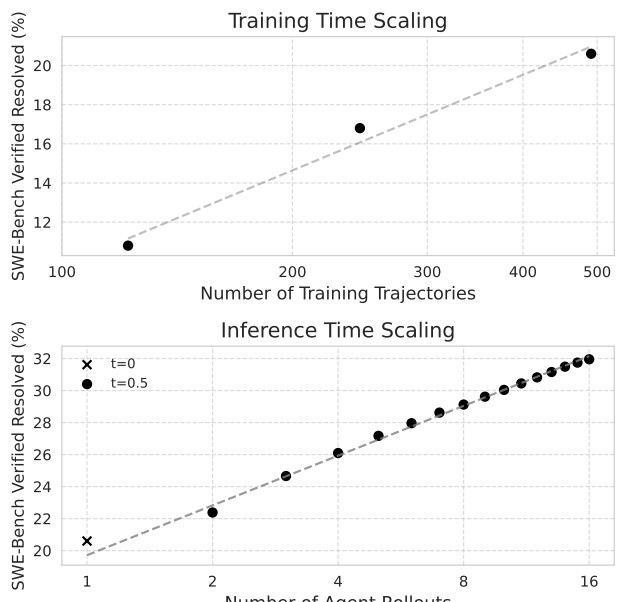

Figure 1: SWE-Gym enables scalable improvements for software engineering agents. **Top**: Scaling the amount of training data shows consistent performance improvements as we obtain more training trajectories, with no signs of saturation at 491 trajectories. We use temperature $t = 0$ for evaluation. **Bottom**: For inference time scaling, we generate a number of candidate trajectories per task and select the best using a verifier trained on SWE-Gym. This approach demonstrates roughly log-linear gains with the number of sampled solutions. $t = 0$ (excluded from regression) is used as the first hypothesis to be consistent with the top figure; later rollouts use $t = 0.5$.

ing (Shao et al., 2024; Yuan et al., 2024), and web navigation (Pan et al., 2024), software engineering currently lacks suitable training environments, and creating environments is uniquely challenging. Real-world software engineering requires interaction with an executable runtime that has been prepared with the appropriate software dependencies and reproducible test suites, among other requirements. These challenges are reflected in the existing resources (Tab. 1). For example, the SWE-Bench (Jimenez et al., 2024) training split contains only solutions (git patches that solve the task), missing the step-by-step actions taken by the developer to create each solution, and executable environments and re-

---
∗ Equal contribution.  † Equal supervision.  [1]UC Berkeley [2]UIUC [3]CMU [4]Apple.  Correspondence to: Jiayi Pan <jiayi-pan@berkeley.edu>, Xingyao Wang <xingyao6@illinois.edu>, Alane Suhr <suhr@berkeley.edu>, Yizhe Zhang <yizzhang@apple.com>.

*Proceedings of the 42^{nd} International Conference on Machine Learning*, Vancouver, Canada. PMLR 267, 2025. Copyright 2025 by the author(s).

ward signals. R2E (Jain et al., 2024) uses synthetic tasks that are very far from real-world problems, while datasets such as APPS (Hendrycks et al., 2021a) focus only on isolated tasks rather than realistic repository-level coding problems.

To bridge this gap, we present SWE-Gym, the **first training environment** combining real-world software engineering tasks from GitHub issues with pre-installed dependencies and executable test verification. SWE-Gym contains 2,438 Python tasks sourced from 11 popular open-source repositories (Tab. 2), providing useful environments for training LMs as agents and verifiers.

**SWE-Gym supports training state-of-the-art open-weight SWE agents**. Based on the OpenHands (Wang et al., 2024c) agent scaffold for general-purpose software development (§2), we fine-tune a 32B Qwen-2.5 coder model (Hui et al., 2024b) using only 491 agent-environment interaction trajectories sampled using SWE-Gym, and achieve substantial absolute improvements of +12.3% (to 15.3%) and +13.6% (to 20.6%) in resolution rate on SWE-Bench Lite and SWE-Bench Verified respectively (§4.2).

**SWE-Gym is effective across agent scaffolds**. In another agent scaffold based on a specialized workflow (Moat-lessTools; Örwall 2024; §2), we experiment with self-improvement, where the LM interacts with SWE-Gym, receives reward from it, and learns to improve itself through rejection sampling fine-tuning. This self-improvement boosts performance up to 19.7% on SWE-Bench Lite.

**SWE-Gym supports training verifier models to enable inference-time scaling**. We use test suites included in SWE-Gym to determine whether sampled agent trajectories are successful or not. Given these samples, we train a verifier model (i.e., an outcome-supervised reward model; Cobbe et al., 2021) that estimates a trajectory's probability of success. This enables inference-time scaling, where we sample multiple agent trajectories and select the one with the highest estimated reward according to the verifier. This further improves the resolution rate to 32.0% (+11.4% absolute improvement) on SWE-Bench Verified (§5.1.1; Fig. 1 bottom) and 26.0% on SWE-Bench Lite (§5.1.2), establishing a new state-of-the-art among systems with publicly accessible weights (Tab. 9).

**Our baseline training and inference-time scaling methods on SWE-Gym yield continuously improved results with increasing compute** (Fig. 1). In the training phase, performance scales with the number of sampled trajectories up to our current limit of 491 trajectories, suggesting that performance is currently limited by the compute budget for sampling rather than the number of tasks in SWE-Gym. Similarly, using the agent and verifier trained by SWE-Gym, the bottom panel shows that using more compute during inference time steadily improves the performance.

## 2. Related Work

**Agents that solve GitHub issues.** We focus on software engineering agents designed to automatically resolve GitHub issues within the SWE-Bench framework (Jimenez et al., 2024). These agents take a GitHub issue and its associated code repository as input and generate a valid code modification (i.e., a git diff patch) to address the issue. The correctness of these modifications is verified using a human-written test suite. Existing agent designs are categorized by the extent of human priors integrated into their workflows: **Specialized workflows** (Xia et al., 2024; Örwall, 2024; Zhang et al., 2024b; Chen et al., 2024) involve human-defined stages (e.g., localization, code editing, patch re-ranking), where a LM is iteratively prompted for each stage to produce the final result. This approach reduces the task horizon and minimizes the need for long-term planning. However, specialized workflows require significant human engineering, may not generalize to novel issue types, and can fail if intermediate steps encounter problems. In contrast, **general-purpose prompting** ((Yang et al., 2024; Wang et al., 2024c)) rely on LM's ability to plan over long horizons and generate actions based on a history of interactions without heavily pre-defined workflows. While more flexible, general approaches demand higher capabilities from the underlying LM and can be computationally expensive due to multiple interaction rounds. The most successful existing SWE agents are built on proprietary language models like GPT-4 or Claude and utilize specialized workflows to overcome these models' limitations. This contrasts with other sequential decision-making domains (Silver et al., 2017; Akkaya et al., 2019), where learning-based approaches, such as reinforcement learning, drive success by enabling systems to learn from interactions and rewards to develop task competence. A key barrier in the SWE agent domain is the lack of appropriate training environments. Our experiments show that SWE-Gym can be used to build strong learning-based agents, accelerating research in this area.

**Environments for training software agents.** There is no existing dataset suitable for training software engineering agents. SWE-Bench (Jimenez et al., 2024) is widely used for evaluating software engineering performance, but its training split lacks executable environments and success signals present in the evaluation split, making it useful only for imitation learning approaches. HumanEval (Chen et al., 2021) is designed for standalone code generation tasks, akin to coding competitions. Therefore, it falls short of addressing the complex challenges inherent in real-world, repository-level software engineering tasks, which involve thousands of files, millions of lines of code, and tasks such as bug fixing, feature development, and system optimization. Similarly, R2E (Jain et al., 2024) is a small evaluation dataset

Table 1: SWE-Gym is the first publicly available training environment combining real-world SWE tasks from GitHub issues with pre-installed dependencies and executable test verification. *Repository-level*: whether each task is situated in a sophisticated repository; *Executable Environment*: whether each task instance comes with an executable environment with all relevant dependencies pre-installed; *Real task*: whether task instruction is collected from human developers.

| Dataset (split) | Repository-Level | Executable Environment | Real task | # Instances (total) | # Instances (train) |
|---|:---:|:---:|:---:|---:|---:|
| CodeFeedback (Zheng et al., 2024b) | ✗ | ✗ | ✓ | 66,383 | 66,383 |
| APPS (Hendrycks et al., 2021a) | ✗ | ✓ | ✓ | 10,000 | 5,000 |
| HumanEval (Chen et al., 2021) | ✗ | ✓ | ✓ | 164 | 0 |
| MBPP (Tao et al., 2024) | ✗ | ✓ | ✓ | 974 | 374 |
| R2E (Jain et al., 2024) | ✓ | ✓ | ✗ | 246 | 0 |
| SWE-Bench (train) (Jimenez et al., 2024) | ✓ | ✗ | ✓ | 19,008 | 19,008 |
| SWE-Gym Raw | ✓ | ✗ | ✓ | 64,689 | 64,689 |
| SWE-Bench (test) (Jimenez et al., 2024) | ✓ | ✓ | ✓ | 2,294 | 0 |
| SWE-Gym | ✓ | ✓ | ✓ | 2,438 | 2,438 |

with 246 instances and, due to its synthetic nature, lacks the realism and complexity in real-world software engineering scenario. Our proposed SWE-Gym instead uses real-world GitHub issues as task, and associated executable unit tests for evaluation. This results in realistic and complex task formulations, aligning closely with real-world challenges.

**Post-training: From chatbots and reasoners to agents.** Post-training, which fine-tunes pre-trained language models using supervised or reinforcement learning, significantly improves model performance across various domains. Techniques like RLHF (Ouyang et al., 2022) have become standard for adapting language models into chatbots, improving both performance and alignment (Qwen Team, 2024). In math reasoning, datasets such as MATH (Hendrycks et al., 2021b) and GSM-8K (Cobbe et al., 2021) facilitate the training and evaluation of policy and verifier models (Cobbe et al., 2021; Wang et al., 2024a). Earlier works (Wang et al., 2024b; Chen et al., 2023; Zeng et al., 2023; Wu et al., 2024) demonstrate that distilling agent trajectories from stronger models improve weaker models. Recent studies (Xi et al., 2024; Zhai et al., 2024; Bai et al., 2024) explore self-improving methods, showing that reinforcement learning or rejection sampling fine-tuning guided by reward enables LMs to enhance themselves without more capable teachers.

However, post-training typically depends on expert demonstration data or training environments with reliable reward signals, which are largely absent in the software engineering domain. This has led to a reliance on prompting-based methods with proprietary language models. Our work addresses this gap with SWE-Gym, a training environment based on real-world software engineering tasks that uses expert-written tests as reward signals. Our experiments demonstrate that SWE-Gym can build strong SWE agents without prompt engineering.

## 3. SWE-Gym Environment

SWE-Gym comprises 2,438 real-world software engineering tasks sourced from pull requests in 11 popular Python repositories, with pre-configured executable environments and expert-validated test cases, constructed in close alignment with SWE-Bench (Jimenez et al., 2024). These repositories are separate from those used in SWE-Bench to avoid contamination. These tasks require SWE agents to develop test-passing solutions for real-world GitHub issues using provided codebases and executable environments. Such agents must map from natural language descriptions of the issue, as well as the initial state of the repository, to a pull request represented as a git patch.

We also identify a subset of 230 tasks, SWE-Gym Lite, which contains generally easier and more self-contained tasks that are suitable for rapid prototyping, in alignment with SWE-Bench Lite (Jimenez et al., 2024). To support future research in SWE agent development and automatic dataset synthesis, we also release SWE-Gym Raw, a large set of Python GitHub issues without executable environments (64,689 instances spanning 358 Python repositories).

### 3.1. Dataset Construction

**Identify Repositories.** We first use SEART GitHub search[1] to filter a list of initial repositories. Unlike SWE-Bench, which focuses on the top 5k most downloaded PyPI libraries (Jimenez et al., 2024), we select Python repositories that were created before July 1, 2022 and have more than 500 stars, with at least 300 lines of code, more than 500 pull requests (PRs) and 100 contributors. This results in 358 repositories.

**Extracting Training Instances from Repositories.** We use SWE-Bench's instance extraction script to convert these repositories into task instances, each corresponding to a

---

[1] https://seart-ghs.si.usi.ch/

| Category | Metric | SWE-Gym | SWE-Gym Lite |
|---|---|---|---|
| Size | # Instances | 2,438 (2,294) | 230 (300) |
| | # Repos | 11 (12) | 11 (12) |
| Issue Text | Length by Words | 239.8 (195.1) | 186.2 (175.9) |
| Codebase | # Non-test Files | 971.2 (2944.2) | 818.8 (2988.5) |
| | # Non-test Lines | 340675.0 (363728.4) | 340626.2 (377562.4) |
| Gold Patch | # Lines edited | 69.8 (32.8) | 10.6 (10.1) |
| | # Files edited | 2.5 (1.7) | 1.0 (1.0) |
| | # Func. edited | 4.1 (3.0) | 1.4 (1.34) |
| Tests | # Fail to Pass | 10.0 (9.0) | 2.04 (3.5) |
| | # Total | 760.8 (132.5) | 99.9 (85.2) |

Table 2: Statistics comparing SWE-Gym with the SWE-Bench test split (in parenthesis). Except for size metrics, we report the average value across instances.

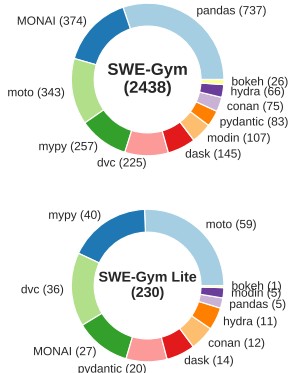

Figure 2: Repository distribution of SWE-Gym instances.

GitHub issue including the natural language description of the issue, a snapshot of the repository in which the issue was created, and a set of unit tests. Over the 358 repositories, we extract 64,689 task instances. We refer to this dataset as SWE-Gym Raw, which is over three times larger than the 19k instances gathered in previous work (Jimenez et al., 2024) and includes nearly ten times as many repositories.

While SWE-Gym Raw instances contain code, issue descriptions, and the solution, they do not contain executable environments or a guarantee that its unit tests are effective in evaluating the correctness of a solution. Thus, we focus on 11 repositories with numerous instances and semi-manually create executable environments for them.

**Version Training Instances.** Associating instances with their respective version numbers (e.g. `1.2.3`) and setting up environments version-by-version makes the environment collection process more practical by avoiding redundant setup work. We generalize SWE-Bench's versioning script to support versioning via script execution, and semi-automatically collect versions for each instance based on information available in the repository (e.g., `pyproject.toml`, git tag, etc).

**Setup Executable Environments and Verify Instances.** Creating executable environments with pre-installed dependencies is crucial for developing software engineering agents, as it mirrors deployment settings and allows for incremental unit test feedback. Configuring dependencies for specific codebase versions is challenging due to the lack of a universal Python package installation method and backward compatibility issues, especially for older GitHub issues. Ignoring these environments could introduce distribution bias, diminishing SWE-Gym's utility. To address this, we manually configure dependencies for each task instance using relevant configuration files (e.g., `requirements.txt`), CI scripts, or documentation from the repository snapshot at the time of issue creation. We then use SWE-Bench's

execution-based validation script to ensure that the gold patch (the human-submitted code diff) passes more unit tests than the original code. This process required approximately 200 human annotation hours[2] and 10,000 CPU core hours. After validation and filtering out failed instances, we obtained 2,438 unit-test-validated instances from 11 repositories. For full reproducibility, we publicly release pre-built Docker images for each instance, totaling 6 TB.

### 3.2. SWE-Gym Lite

Solving software engineering tasks is computationally intensive, costing usually $1 or more per task with frontier models (Wang et al., 2024c). To improve research efficiency via faster agent evaluation, Jimenez et al. (2024) introduce SWE-Bench Lite, a canonical subset of 300 instances from SWE-Bench. Following the SWE-Bench Lite filtering pipeline,[3] we delineate the **SWE-Gym Lite** split, comprising 230 instances. Similar to SWE-Bench Lite, this subset excludes tasks that require editing more than one file, tasks with poorly described problem statements, those with excessively complex ground-truth code diffs, and tests focused on error message validation.

### 3.3. Dataset Statistics

Fig. 2 illustrates that the task distribution across repositories exhibits a long-tail pattern. Notably, tasks associated with `pandas` comprise nearly one-third of the total, whereas tasks related to `bokeh` represent a mere one percent.

Our analysis suggests that tasks in SWE-Gym are on average harder than those included in SWE-Bench. Tab. 2 shows that SWE-Gym has statistics similar to SWE-Bench, with several key differences. Codebases in SWE-Gym, on average, have relatively fewer files than SWE-Bench, but a

---

[2]Annotations are done by a subset of the authors.

[3]For details on its construction process, see `https://www.swebench.com/lite.html`.

similar number of total lines of code. However, gold patches in SWE-Gym have significantly more lines and files edited when compared to SWE-Bench's gold patches. Additionally, we find models have consistently lower performance on SWE-Gym compared to SWE-Bench.[4] Beyond models and scaffolds overfitting to SWE-Bench, the decreased performance on SWE-Gym may also be due to our inclusion of sophisticated repositories like `pandas` and `MONAI`.

## 4. Training LMs as Agents with SWE-Gym

We experiment with training language model agents using SWE-Gym. We use two agent scaffolds (OpenHands, Wang et al. 2024c, §4.2; Moatless Tools, Örwall 2024, §4.3).

### 4.1. Setting

**Agent Scaffolds.** Recent LM-based SWE agents comprise a base language model, and a set of tools and prompts this base model has access to. This set of tools and prompting strategies is referred to as an agent scaffold, and recent work has developed numerous scaffolds for different purposes (refer to §2 for examples). We experiment with two types of agent scaffolds: one for general-purpose prompting (OpenHands CodeAct; Wang et al. 2024c) and one for specialized workflows (MoatlessTools; Örwall 2024), which allows us to measure the efficacy of SWE-Gym across diverse deployment settings.

**Policy Improvement Algorithm.** We use SWE-Gym to improve the underlying LM for a given SWE agent. As a baseline, we employ a simple policy improvement algorithm: rejection sampling fine-tuning (a.k.a. filtered behavior cloning), where we fine-tune the base LM on *success* trajectories sampled from SWE-Gym.

**Evaluation Metrics.** We use the standard SWE agent benchmarks SWE-Bench Lite and Verified (Jimenez et al., 2024) for evaluation. We report (1) **resolution rate (%)**, the proportion of resolved task instances, and (2) **Empty Patch (%)**, the proportion of trajectories where none of the code in the repository is edited. We use OpenHands remote runtime (Neubig & Wang, 2024) to parallelize evaluation (e.g., execute unit tests).

**Technical Details.** For base LMs, we use `Qwen-2.5-Coder-Instruct` (Hui et al., 2024a) 7B, 14B, and 32B. §B.2 contains training run details.

### 4.2. Training General-Purpose Prompting Agents

In this section, we use OpenHands (version CodeActAgent 2.1, Wang et al. 2024b;c) as our agent scaffold, which is based on general-purpose ReAct-style prompting (Yao et al.,

2023). In contrast to specialized-workflows-agents (§2), it relies on the LM to generate actions and do planning. It equips the base LM with a bash terminal and a file editor. We disable the browser feature of OpenHands in this work.

**Trajectory Collection.** By rejection sampling, we obtain 491 successful trajectories from SWE-Gym,. These trajectories are sampled from `gpt-4o-2024-08-06` and `claude-3-5-sonnet-20241022` with different temperature settings. Each successful trajectory, on average, has roughly 19 turns and approximately 19,000tokens.[5] Although SWE-Gym offers many more tasks and allows repeated sampling, our 491 trajectories are limited primarily by computational budget.

**Training on SWE-Gym trajectories turns LM into effective agents to fix issues.** As shown in Tab. 3, the pretrained base model achieves resolution rates of 3.0% and 7.0% on SWE-Bench Lite and Verified, respectively. After fine-tuning on 491 trajectories[6], it improves by up to 12.3% (3.0% → 15.3%) and 13.6% (7.0% → 20.6%).

**Training reduces stuck-in-loop behavior.** For agent tasks, open-weight LMs often get stuck in loops, where the model perpetually generates the same action for multiple turns, especially when prompted with general-purpose prompts (§2). Thus, we report **Stuck in Loop (%)**, the percentage of trajectories where the agent repeats the same action three times consecutively. As shown in Tab. 3, zero-shot pretrained models often get stuck in loops; even the largest 32B model is trapped in 29.4% of SWE-Bench Verified tasks. Fine-tuning on trajectories from SWE-Gym consistently reduces the stuck-in-loop rate by 4.6–18.6% across both SWE-Bench Lite and Verified tasks, except for the 32B model on SWE-Bench Lite, which increases by 1.5% due to its already low loop rate. This coincides with a decrease in the empty patch rate, likely enabling the agent to perform more code edits.

**Performance scales with model size.** Rather unsurprisingly, larger base models consistently improve the resolution rate, empty patch rate, and stuck-in-loop rate (Tab. 3).

**Self-improvement remains ineffective.** In addition to fine-tuning on trajectories sampled from strong teacher models, we also experiment with fine-tuning on trajectories sampled directly from the policy being updated. We use the fine-tuned 32B model to sample 6 trajectories per SWE-Gym instance (using temperature $t = 0.5$), obtaining 868 successful trajectories (i.e., on-policy trajectories). We further fine-tune the base 32B model on a mixture of 868 on-policy trajectories and the previously collected 491 off-policy trajectories. When evaluating this fine-tuned model on SWE-Bench Lite, we observe the resolution rate drop from 15.3

---

[4]§B.4 contains details of these experiments.

[5]Tab. 8 contains more statistics of the sampled trajectories.
[6]We use a sampling temperature of 0 unless otherwise specified.

Table 3: Model performance (fine-tuned on 491 SWE-Gym-sampled trajectories) on SWE-Bench (Jimenez et al., 2024) using OpenHands (Wang et al., 2024c) as agent scaffold. We use `Qwen-2.5-Coder-Instruct` as the base model.

| Model | Empty Patch (%, ↓) | | | Stuck in Loop (%, ↓) | | | Avg. Turn(s) | | | Resolve Rate (%, ↑) | | |
| Size | zero-shot | fine-tuned | Δ | zero-shot | fine-tuned | Δ | zero-shot | fine-tuned | Δ | zero-shot | fine-tuned | Δ |
|---|---|---|---|---|---|---|---|---|---|---|---|---|
| | | | | | *SWE-Bench Lite (300 instances)* | | | | | | | |
| 7B | 40.3 | 29.7 | -10.7 | 47.0 | 31.0 | -16.0 | 20.3 | 22.2 | +1.9 | 1.0 (± 1.0) | 10.0 (± 2.4) | +9.0 |
| 14B | 49.7 | **18.1** | -31.6 | 31.7 | 27.1 | -4.6 | 23.2 | 21.4 | -1.8 | 2.7 (± 1.9) | 12.7 (± 2.3) | +10.0 |
| 32B | **27.0** | **18.1** | -8.9 | **16.7** | **18.1** | +1.5 | 15.5 | 29.3 | +13.9 | **3.0** (± 1.4) | **15.3** (± 2.5) | **+12.3** |
| | | | | | *SWE-Bench Verified (500 instances)* | | | | | | | |
| 7B | 45.8 | 33.8 | -12.0 | 39.6 | **21.0** | -18.6 | 21.9 | 35.3 | +13.4 | 1.8 (± 1.1) | 10.6 (± 2.1) | +8.8 |
| 14B | 44.9 | 14.5 | -30.4 | 32.1 | 21.3 | -10.7 | 25.5 | 30.1 | +4.6 | 4.0 (± 1.6) | 16.4 (± 2.0) | +12.4 |
| 32B | **9.5** | 13.8 | +4.3 | **29.4** | 23.8 | -5.6 | 24.6 | 31.6 | +7.0 | **7.0** (± 1.3) | **20.6** (± 2.1) | **+13.6** |

to 8.7%, suggesting that self-improvement is not yet working. We hypothesize that we could achieve improved results using more advanced policy optimization methods, such as proximal policy optimization (PPO) (Schulman et al., 2017), or with a stronger base model. These directions remain promising avenues for future investigation.

### 4.3. Self-Improvement with Specialized Workflow

Unlike OpenHands, which offers freedom in long-horizon planning, MoatlessTools constrains the language model's action space to pre-defined specialized workflows, reducing task horizons. Specialized workflows outperform general-purpose prompting for open-weight LMs. In Tab. 3 and Tab. 4, the 7B and 32B LM achieve zero-shot resolution rates of 7% and 19% with MoatlessTools, compared to 1.0% and 3.0% with OpenHands on SWE-Bench Lite.

Given MoatlessTools' improved zero-shot performance and shorter task horizon, we hypothesize that self-improvement without a strong teacher is achievable using this scaffold and training on SWE-Gym. With a limited compute budget, we conduct this experiment with only 7B and 32B models, using LoRA (Hu et al., 2022) for the 32B model for improved efficiency. We use the 7B model for ablation experiments.

We use iterative rejection sampling fine-tuning for policy improvement. Each iteration involves (a) performing 30 high-temperature (1.0) rollouts per task on SWE-Gym-Lite and adding successful trajectories to the fine-tuning dataset, and (b) fine-tuning the policy on these filtered trajectories. After two iterations, further improvements are negligible.

**Data Bias Impacts Performance.** Repeated sampling, as in Brown et al. (2024), shows that task success probability follows a long-tail distribution (Fig. 6), where more samples increase solved instances. While broader task coverage benefits training, it introduces a bias toward easier tasks, making it suboptimal to train on all successful trajectories, as first observed in math reasoning Tong et al. (2024).

**Mitigating Bias with Per-Instance Capping.** We introduce per-instance capping—a method that limits the maximum number of selected samples per task. As illustrated in Fig. 6,

Table 4: resolution rate (RR) and Empty patch rate (EP) on SWE-Bench Lite with the MoatlessTools Scaffold after online rejection sampling fine-tuning (temperature $t = 0$).

| Setting | 7B Model | | 32B Model | |
| | EP(%, ↓) | RR(%, ↑) | EP(%, ↓) | RR(%, ↑) |
|---|---|---|---|---|
| Zero-Shot | 56.3% | 7.0% | 24.3% | 19.0% |
| Iteration 1 | 29.0% | 9.0% | 18.3% | **19.7%** |
| Iteration 2 | **23.3%** | **10.0%** | **9.7%** | **19.7%** |

this balances dataset bias and size. A low cap reduces dataset size and performance (§5.2), while a high cap skews the distribution toward easier tasks. Empirically, a threshold of 2 achieves a good balance, slightly outperforming the full dataset and improving training speed (Tab. 6). We rank trajectories by the number of model response rounds required, preferring fewer.

**Results.** Results. After two policy improvement iterations (Tab. 4), the 7B model's resolution rate increased from 7.0% to 9.0% after the first iteration and to 10.0% after the second. In contrast, the 32B model improved from 19.0% to 19.7% after the first iteration with no further gains. We attribute the limited gains in the 32B model to the scaffold's restricted action space and the rejection sampling fine-tuning method.

## 5. Scaling Agent Performance with SWE-Gym

We explore two scaling directions enabled by SWE-Gym to enhance agent performance: inference-time scaling (§5.1) and training-time data scaling (§5.2).

### 5.1. Inference-Time Scaling with Verifiers

Trajectories sampled from SWE-Gym can be used not only for training a policy, but also for training a verifier (i.e., reward) model. We train an outcome-supervised reward model (ORM) (Cobbe et al., 2021) that, given the relevant context of the task execution (including the problem statement, agent trajectory, and current git diff), generates a score that estimates the probability that the agent has solved the problem. We experiment with using this model to rerank

candidate trajectories sampled from a SWE agent policy, and show that such learned verifiers enable effective inference-time scaling for further performance improvement.

### 5.1.1. VERIFIER FOR GENERAL-PURPOSE PROMPTING

For OpenHands agents (Wang et al., 2024b;c) with general-purpose prompting (§2), we train a verifier (ORM) that takes as input the trajectory $\tau = [o_1, a_1, o_2, a_2, \ldots, o_n, a_n]$, represented as an interleaved sequence of observations and actions, and generates a scalar reward $r \in [0, 1]$. Observations $o_k$ include the task problem statement, command execution output, error messages, etc; action $a_k$ can be bash command or file operations (e.g., edit, view) from the agent.

**Training and Inference.** We fine-tune 32B `Qwen2.5-Coder-Instruct` to label trajectories as successful or unsuccessful using output tokens `<YES>` and `<NO>` respectively.[7] For training data, we re-use two sets of trajectories we sampled on SWE-Gym for agent training in §4.2: (1) **off-policy trajectories** which contain 443 successful trajectories; (2) **on-policy trajectories** which contain 875 successful trajectories sampled from the fine-tuned `Qwen2.5-Coder-Instruct-32B`.[8] We combine both on-policy and off-policy trajectories, randomly sample the same amount of unsuccessful trajectories from each subset (1,318 each), and combine them as our dataset for verifier training (total 2,636 trajectories). We fine-tune the model to predict `<YES>` for successful trajectories and `<NO>` for unsuccessful ones.

At inference time, conditioned on the prompt and the agent trajectory $\tau$, we use SGLang (Zheng et al., 2024a) to obtain the log probability of the next token being `<YES>` ($l_y$) or `<NO>` ($l_n$). We then calculate the reward as the probability of success by normalizing the log probability: $r = \exp(l_y)/(\exp(l_y) + \exp(l_n))$.

**Metrics.** We report two metrics: **(1) Pass@$k$**, the proportion of tasks with at least one successful solution among $k$ samples, and **(2) Best@$k$**, the success rate of the highest-reward trajectories selected by the verifier from $k$ samples per task. Pass@$k$ measures solution discovery (upper bound for Best@$k$); Best@$k$ evaluates verifier accuracy. Mean and variance calculation are detailed in §B.1, following Lightman et al. (2023).

**Results.** Fig. 3 shows how Pass@$k$ and Best@K scale with the number of sampled agent trajectories using the fine-tuned 32B model as the agent model. Pass@$k$ demonstrates strong improvement, rising from 20.6 to 37.8% resolution rate as $k$ increases from 1 to 8, and up to 42.8@$k$=16. The Best@$k$ metric, which relies on our verifier's ability to se-

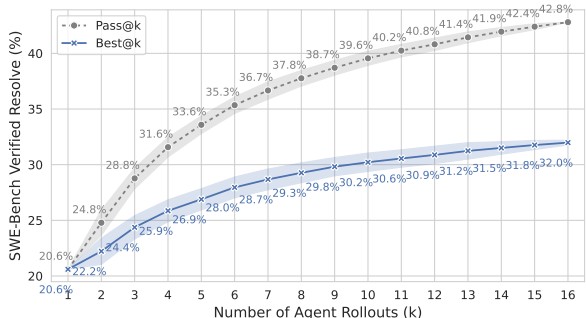

Figure 3: Increasing inference-time compute improves performance on SWE-Bench Verified with a learnt verifier. Both the agent and the verifier are a `Qwen2.5-Coder-Instruct-32B` model fine-tuned on the corresponding dataset (§5.1.1). OpenHands is used as the agent scaffold.

lect the best trajectory, demonstrates more modest but steady progress, improving from a resolution rate of 20.6@1 to 29.8@8, and up to 32.0@16. The gap between Pass@$k$ and Best@$k$, due to the imperfect performance of our trained verifier, indicates there is room for improvements in reward modeling for coding agents. Surprisingly, we found that fine-tuning the verifier model using LoRA (Hu et al., 2022) (29.8@8) with Unsloth (Unsloth Team, 2024) performs better than full-parameter fine-tuning (27.2@8), potentially due regularization. Furthermore, as shown in Fig. 1 (bottom), the Best@$k$ curve exhibits strong linearity on a logarithmic scale, indicating a promising scaling behavior.

**Training data matters for verifier.** We experiment with variations on the choice of training data for our verifier model. Using full-parameter fine-tuning on `Qwen-2.5-Coder-Instruct-32B`, we use different mixtures of on- and off-policy trajectories, as well as different distributions of successful and unsuccessful trajectories. As shown in Fig. 8, our ablation study demonstrates that the choice of training data can significantly impact verifier performance. Training with a mixture of off-policy and on-policy data yields the best results (our default setting), reaching a resolution rate of 27@8. In contrast, using only on-policy data from the fine-tuned model shows moderate but limited improvement, while training exclusively on off-policy data from Claude and GPT leads to early performance plateaus around 22% resolution rate. Our findings indicate that verifier training benefits most from a diverse dataset combining both off-policy and on-policy examples.

### 5.1.2. VERIFIER FOR SPECIALIZED WORKFLOW

For MoatlessTools agents with specialized workflows, given that it doesn't have a turn-taking action-observation trajectory like OpenHands CodeActAgent, we prepare verifier

---

[7]§B.6 includes the verifier prompt template.

[8]We keep only trajectories within 32k-token length for training, which may reduce their number compared to Section 4.2.

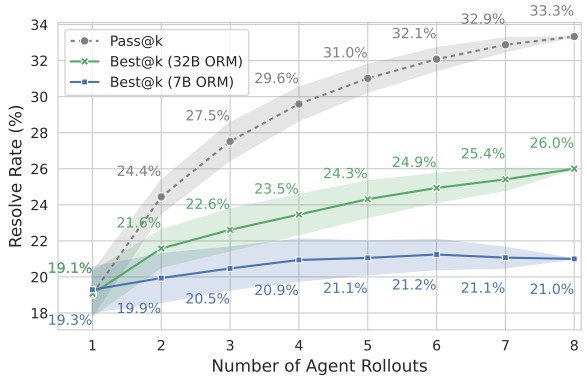

Figure 4: Scaling inference-time compute for MoatlessTools Agents (32B) with learned verifiers on SWE-Bench Lite. Temperature $t = 0.5$.

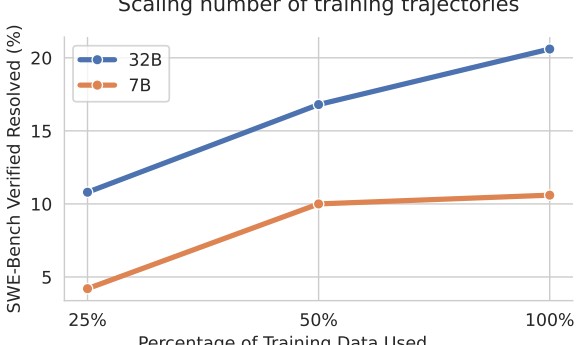

Figure 5: Scaling effects of increasing the number of randomly sampled trajectories for training.

inputs through a parsing process adopted from Zhang et al. (2024a), which combines task descriptions, relevant agent context, and generated patches.[9] We train the verifier to map from this input to a single token indicating task success.

Following the training procedure described in §5.1.1, we train 7B and 32B verifiers using on-policy trajectories from the last (2nd round of sampling, applying LoRA (Hu et al., 2022). To address the easy-data bias in the training dataset, we cap the number of successful trajectories per instance at two and balance the data by subsampling failure cases to match the same number of successful ones.

**Results.** We evaluate the verifiers by sampling from an agent policy with $k = 8$ at temperature 0.5. As shown in Fig. 4 and Fig. 7, these verifiers enable effective scaling across verifier and policy sizes: the 7B verifier improves from 10 to 13.3% resolution rate on SWE-Bench Lite when paired with a 7B policy, while the 32B verifier improves from 19.7 to 26.3% when paired with a 32B policy. The 7B verifier plateaus after $k = 4$ samples when ranking trajectories from both 7B and 32B agents. In contrast, the 32B verifier continues improving even at $k = 8$, suggesting that verifier size significantly affects scaling behavior.

### 5.2. Training-Time Scaling with Data

We then examine how scaling the amount of training data affects agent performance using 491 sampled trajectories from §4.2. We simulate three scaling methods through subsampling: (1) **Scaling trajectories**, where trajectories are randomly dropped (Fig. 5); (2) **Scaling unique task instances**, where only one successful trajectory per task instance is selected (Fig. 9); and (3) **Scaling repositories**, which sequentially includes all instances from each repository to assess repository-level diversity.

**Setup.** Using OpenHands (Wang et al., 2024c) and the fine-tuning approach described in §4.2, we evaluate these scaling approaches on SWE-Bench Verified: scaling the number of trajectories, by subsampling from the full trajectory dataset from §4.2 (at most 491 trajectories); unique instance scaling on these trajectories deduplicated by instance ID (at most 294 trajectories), and repository-based scaling where we sort repositories alphabetically and include all trajectories from each repository in order (e.g., first 25% contains complete trajectories from the first N repositories). We compare models trained on 25%, 50%, and 100% of the full dataset for each approach, sampling training subsets using the methods described above for each scaling approach.[10]

**Scaling trends suggest instance and repository diversity is not yet a bottleneck.** Fig. 5 demonstrates substantial scaling behavior, with consistent improvements in resolution rate as the number of training trajectories randomly increases, particularly for the 32B model. These results suggest that SWE-Gym's current size and repository diversity are likely not performance bottlenecks - further improvements could likely be achieved by allocating more computing resources to sampling more training trajectories.

Fig. 9 reveals comparable overall performance between different scaling approaches up to where deduplication takes effect. While Random Scaling (No Dedup.) achieves higher final performance, this is likely due to having more trajectories (491 vs 294) rather than better scaling efficiency. Among deduplicated approaches, Repository Scaling shows stronger initial performance at 25% data, suggesting that complete repository coverage may provide more coherent learning signals early in training. These results suggest that the repository and instance diversity of SWE-Gym is not yet a bottleneck - further improvements could likely be achieved by simply sampling more agent trajectory data for traning, regardless of duplication or repository distribution.

---

[9]We provide the prompt template in §B.5.

[10]Tab. 7 contains detailed statistics of these datasets.

# 6. Conclusions, Limitations, and Future Work

In this paper, we introduce SWE-Gym, the first training environment that addresses critical gaps in enabling scalable learning for software engineering agents. By combining real-world Python tasks with repository-level context, pre-configured execution environments, and test verifications, SWE-Gym will be a foundation for advancing LM agent training research. Through extensive experiments, we demonstrate that SWE-Gym enables both agent and verifier models to achieve significant improvements in resolving complex software tasks. Our findings highlight the scalability of these approaches, revealing potential for continuous performance gains with increased compute.

We see many research directions that we are excited to explore in the future:

1. **Automatic Environment Synthesis** SWE-Gym, while effective, is limited by its environment diversity, including the number of repositories, types of tasks, and programming languages. We view environment synthesis—via automated environment creation, test-case generation, or task generation—as a critical next step.

2. **Self-Improvement with Reinforcement Learning** Despite notable progress, our self-improvement results are modest. Training language model agents with large-scale online reinforcement learning is a promising direction for further improvements.

3. **Human-Agent Interaction** Current SWE settings focus solely on task completion, neglecting human-in-the-loop collaboration, which is essential for real-world software engineering. Methods like user simulation or learning from offline human-agent interaction data might offer ways for developing collaborative agents that align with human.

# Impact Statement

This work presents SWE-Gym, an environment for training software engineering agents, with strong empirical results on its effectiveness. We discuss a few important societal implications to consider. First, improving automated software engineering capabilities could increase developer's productivity and accessibility across industries. Although current models are primarily research artifacts and not yet production-ready, they can support critical open-source infrastructure and potentially make software development more accessible. Secondly, as these agents become more capable, they may impact software engineering jobs and require careful consideration around code ownership, licensing, and attribution. Additionally, while we focus on legitimate software engineering tasks, similar techniques could potentially be misused to automate the creation of malicious code. We encourage future work to further explore frameworks for responsible deployment of software engineering agents, including considerations around security, safety, and economic impacts.

# Acknowledgments

We thank John Yang and Ofir Press for helpful discussions, and John Yang for assistance in reproducing data analysis results from SWE-Bench. We thank Modal Labs[11] for the GPU compute support through its Academic Credits Program. XW and HJ are partially supported by U.S. DARPA ITM Program No. FA8650-23-C-7316. The views and conclusions contained herein are those of the authors and should not be interpreted as necessarily representing the official policies, either expressed or implied, of DARPA, or the U.S. Government. The U.S. Government is authorized to reproduce and distribute reprints for governmental purposes notwithstanding any copyright annotation therein.

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

# A. Comparison with Concurrent Works

Ma et al. (2024) trains an LM agent, Lingma SWE-GPT, using a method similar to our rejection sampling fine-tuning baseline, with a dataset comparable to our SWE-Gym Raw splits. Without executable unit test feedback, they rely on manually defined heuristics to filter out low-quality trajectories, such as comparing similarity between submitted patches and edit locations with gold patches. The model weights are publicly accessible but not the training pipeline or the dataset.

Most relevant to our work are two consecutive blog posts by Golubev et al. (2024) and Badertdinov et al. (2024), who also construct an executable training environment with real-world tasks from GitHub. Instead of manual configuration, they employ a general environment setup script and simply discard instances that fail the setup process. This approach leads to key differences in dataset size and distribution: while it biases the environment away from tasks with complex dependencies, they successfully collect 6,415 instances, about 1.5 times larger than our dataset. In Golubev et al. (2024), they also study training agents and verifiers with the environment. Additionally, they explore a lookahead setting where a trained verifier ranks and selects the best next action. With a substantially large collection of agent trajectories (80,036 compared to thousands in our experiments) and model size (72B compared to 32B), Their best system achieves 40% accuracy on SWE-Bench Verified. While their dataset and agent trajectories are publicly accessible, the model is not.

In comparison, with a comparable dataset size, our SWE-Gym has executable feedback, avoids potential dataset bias through manual configuration of environments, while providing comprehensive analysis of agent and verifier training, their scaling behaviors, and positive results on agent self-improvement. Our system achieves competitive results with significantly lower compute and a smaller model size (32B vs 72B). Lastly, we open source all artifacts of the project, including dataset, model weights, agent trajectory data and the training pipeline.

| Model | | SWE-Bench | | Openness | |
|---|---|---|---|---|---|
| Name, Model Size | | Lite | Verified | Model | Environment |
| Ma et al. (2024), 72B | | 22.0 | 30.2 | ✓ | ✗ |
| Golubev et al. (2024) Agent and Verifier, 72B | | - | 40.6 | ✗ | ✓ |
| Our SWE-Gym Agent and Verifier, 32B | | 26.0 | 32.0 | ✓ | ✓ |

Table 5: Comparison of model performance on SWE-Bench benchmark and if the model weights and environments are publically accessible (openness).

| Cap | # Traj | Empty Patch ($\%, \downarrow$) | resolution rate ($\%, \uparrow$) |
|---|---|---|---|
| 0 (Zero-shot) | 0 | 56.3 | 7.0 |
| 1 | 36 | 37.3 | 9.0 |
| 2 | 62 | **29.0** | **9.7** |
| 3 | 82 | 43.7 | 7.7 |
| No Cap (All) | 172 | 30.7 | 9.3 |

Table 6: resolution rate and empty patch rate on SWE-Bench Lite with a 7B model trained using different instance capping strategies (Cap).

# B. Experiment Details

### B.1. Mean and Variance for Pass@N and Best@N.

We mostly follow (Lightman et al., 2023) for obtaining the mean and variance for the Pass@N and Best@N curve. Given a total of M rounds of rollouts, for $N < M$, we calculate the mean and variance across 100 randomly selected sub-samples of size $N$ from the $M$ rollouts. For the OpenHands CodeActAgent inference-time scaling curve at §3, we exclude this calculation for N=1 , as we use a temperature of 0 for the first attempt.

### B.2. OpenHands Agent Experiments

During training, we use OpenHands's remote runtime (Neubig & Wang, 2024) feature to execute agent trajectories in parallel on SWE-Gym. We use `torchtune` (PyTorch Team, 2024) for full parameter fine-tuning with a learning rate of `1e-4`, maximum 5 epochs, global batch size of 8, max context length of `32768`. We fine-tuned both 7B, 14B, and 32B variant of the model, and experiments were performed with 2-8x NVIDIA H100 80G GPU on modal (Modal, 2024). The

| | Original | Dedup. | Sorted by Random (Dedup.) | | Sorted by Repo (Dedup.) | |
|---|---|---|---|---|---|---|
| | | | First 25% | First 50% | First 25% | First 50% |
| getmoto/moto | 155 | 72 | 12 | 33 | 0 | 46 |
| Project-MONAI/MONAI | 95 | 53 | 17 | 25 | 53 | 53 |
| pandas-dev/pandas | 70 | 61 | 14 | 30 | 0 | 0 |
| python/mypy | 46 | 27 | 7 | 12 | 0 | 0 |
| dask/dask | 45 | 29 | 8 | 17 | 6 | 29 |
| iterative/dvc | 36 | 24 | 8 | 12 | 0 | 0 |
| conan-io/conan | 20 | 12 | 1 | 7 | 12 | 12 |
| pydantic/pydantic | 11 | 7 | 2 | 4 | 0 | 0 |
| facebookresearch/hydra | 7 | 5 | 2 | 5 | 0 | 5 |
| bokeh/bokeh | 3 | 2 | 1 | 1 | 2 | 2 |
| modin-project/modin | 3 | 2 | 1 | 1 | 0 | 0 |
| **Total** | 491 | 294 | 73 | 147 | 73 | 147 |

Table 7: Distribution of success trajectories used in training-time scaling experiments (§5.2). **Dedup.** denotes that the trajectories are deduplicated by randomly select ONE success trajectory per instance ID; **Sorted by random (repo) X% (Dedup.)** denotes a subset of trajectories taken from the first X% from dedup. instances that are sorted randomly (by repository name).

| | Resolved | Count | Mean | Std | Min | Max | Percentiles | | | | | | |
|---|---|---|---|---|---|---|---|---|---|---|---|---|---|
| | | | | | | | 5% | 10% | 25% | 50% | 75% | 90% | 95% |
| Num. of Messages | ✗ | 5,557.0 | 39.2 | 31.9 | 7.0 | 101.0 | 9.0 | 9.0 | 9.0 | 29.0 | 61.0 | 100.0 | 101.0 |
| | ✓ | 491.0 | 39.9 | 19.9 | 13.0 | 101.0 | 19.0 | 21.0 | 25.0 | 33.0 | 47.5 | 65.0 | 87.0 |
| Num. of Tokens | ✗ | 5,557.0 | 17,218.3 | 17,761.6 | 1,615.0 | 167,834.0 | 1,833.0 | 1,907.0 | 2,268.0 | 12,305.0 | 26,434.0 | 41,182.2 | 51,780.6 |
| | ✓ | 491.0 | 18,578.5 | 11,361.4 | 2,560.0 | 81,245.0 | 5,813.0 | 8,357.0 | 11,559.5 | 15,999.0 | 22,040.5 | 31,632.0 | 39,512.5 |

Table 8: Statistics of SWE-Gym-sampled trajectories. We use the tokenizer from `Qwen-2.5-Coder-Instruct-7B` to estimate the number of tokens.

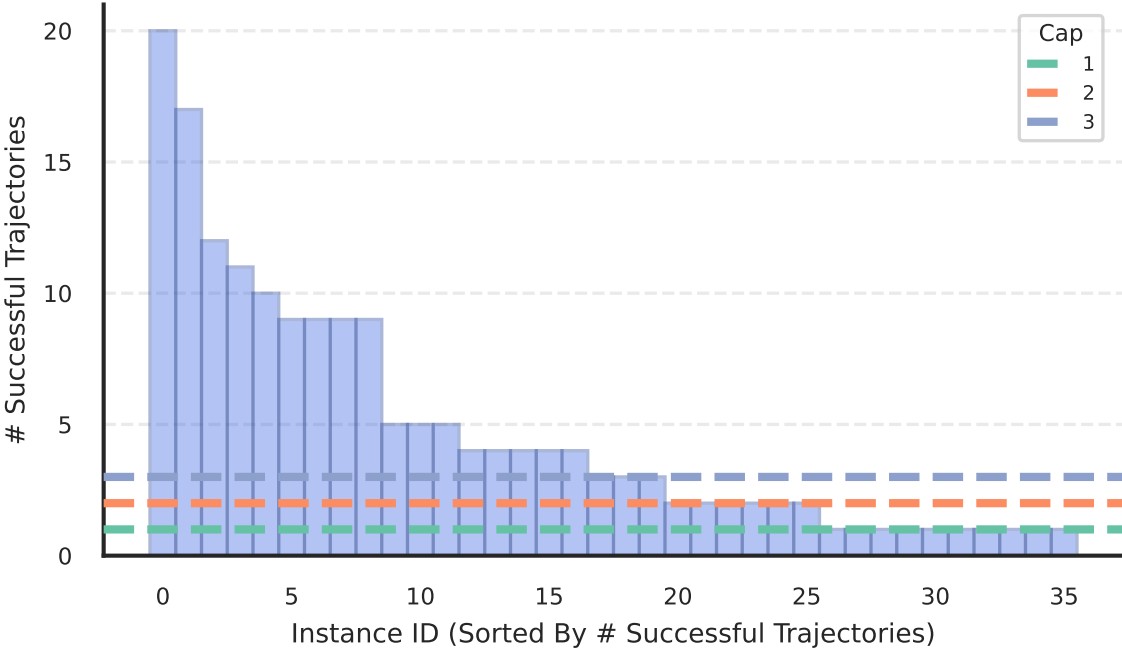

Figure 6: Success distribution over 30 rounds on SWE-Gym Lite with 7B model in zero-shot. The distribution is naturally biased toward easy tasks. Per instance capping reduces this bias but lowers the total trajectory count for training. We set temperature $t = 1$ during sampling.

only exception is in the main experiment of §5.1.1, where we use LoRA (Hu et al., 2022) (29.8% @8) via Unsloth library (Unsloth Team, 2024) to train the verifier for max 2 epochs, while other hyper-parameter stays the same.

Inference during evaluation is bounded by either 100 interaction turns or the base LM's 32k context window length, whichever is reached first.

### B.3. MoatlessTools Agent Experiments

All MoatlessTools models are trained with a context window of `10240`. For experiments with the 7B model, we use torchtune to train the policy model with full-finetuning using 4 H100 GPUs. We set batch size to 8, learning rate to $2 \times 10^{-5}$, and train for 5 epochs.

For the 32B model, we use Unsloth (Unsloth Team, 2024) with a single H100 GPU for LoRA fine-tuning. We set the number of epochs to 5, batch size to 8, LoRA rank to 64, and learning rate to $5 \times 10^{-4}$. We use the same configuration for verifier training.

For MoatlessAgent experiments, we serve the agent with FP8 quantization for improved throughput, which we found to have minimal effects on model performance.

### B.4. Details of OpenHands Trajectory Sampling

As detailed in Tab. 10, we collect a few sets of trajectories for fine-tuning experiments. We collect dataset $D_0$ by sample `gpt-4o-2024-08-06` on SWE-Gym Lite with temperature 0 and collected 19 trajectories that eventually solve the task (evaluated by unit test in SWE-Gym). We then varied the temperatures (setting `t={0.2, 0.3, 0.4, 0.5, 0.8}`) and sample on SWE-Gym Lite. Combining these instances with $D_0$, we get 106 trajectories that solve the given problem ($D_1$). We set the maximum number of turns to be 30 for both $D_0$ and $D_1$. To experiment on the effect of max turn, we set max number of turns to 50 and sample `gpt-4o-2024-08-06` (19 resolved out of 230) and `claude-3-5-sonnet-20241022` (67 resolved out of 230) with temperature 0 on SWE-Gym Lite, and sample `gpt-4o-2024-08-06` (temperature `t={0, 1}`) on SWE-Gym full set (in total 299 resolved out of 4876 instances).

| Agent | Model | Model Size | Training Data | Resolved (%) |
|---|---|---|---|---|
| | | *SWE-Bench Verified (500 instances)* | | |
| RAG | SWE-Llama (Jimenez et al., 2024) | 7B | 10K instances | 1.4 |
| RAG | SWE-Llama (Jimenez et al., 2024) | 13B | 10K instances | 1.2 |
| Lingma Agent (Ma et al., 2024) | Lingma SWE-GPT (v0925) | 7B | 90K PRs from 4K repos | 18.2 |
| Lingma Agent (Ma et al., 2024) | Lingma SWE-GPT (v0925) | 72B | 90K PRs from 4K repos | 28.8 |
| **OpenHands (Wang et al., 2024c) (Ours)** | fine-tuned Qwen2.5-Coder-Instruct | 32B | 491 agent trajectories from 11 repos | 20.6 |
| **OpenHands w/ Verifier (Wang et al., 2024c) (Ours)** | fine-tuned Qwen2.5-Coder-Instruct | 32B (Agent & Verifier) | 491 agent trajectories from 11 repos for agent + $1318 \times 2$ success/failure agent trajectories for verifier | **32.0** |

Table 9: Performance comparison with SWE-Bench (Jimenez et al., 2024) baselines *with publicly accessible weights*. Data source: https://www.swebench.com/, Accessed on Dec 21, 2024.

| Trajectory Set | Sampled from Model | Sampled on Dataset | Temperature | Max Turns | Success trajectories |
|---|---|---|---|---|---|
| $D_0$ | `gpt-4o-2024-08-06` | SWE-Gym Lite | 0 | 30 | 19 (8.26%) |
| | | | **(Cumulative) Total $D_0$** | | **19** |
| | `gpt-4o-2024-08-06` | SWE-Gym Lite | 0.2 | 30 | 11 (4.78%) |
| | `gpt-4o-2024-08-06` | SWE-Gym Lite | 0.3 | 30 | 17 (7.39%) |
| $D_1 \setminus D_0$ | `gpt-4o-2024-08-06` | SWE-Gym Lite | 0.4 | 30 | 21 (9.13%) |
| | `gpt-4o-2024-08-06` | SWE-Gym Lite | 0.5 | 30 | 18 (7.83%) |
| | `gpt-4o-2024-08-06` | SWE-Gym Lite | 0.8 | 30 | 20 (8.70%) |
| | | | **(Cumulative) Total $D_1$** | | **106** |
| | `gpt-4o-2024-08-06` | SWE-Gym Lite | 0 | 50 | 19 (8.26%) |
| $D_2 \setminus D_1$ | `claude-3-5-sonnet-20241022` | SWE-Gym Lite | 0 | 50 | 67 (29.1%) |
| | `gpt-4o-2024-08-06` | SWE-Gym Full | 0 | 50 | *111 (4.55%) |
| | `gpt-4o-2024-08-06` | SWE-Gym Full | 1 | 50 | 188 (7.71%) |
| | | | **(Cumulative) Total $D_2$** | | **491** |

* Run into infrastructure-related error where some instances failed to complete, this number might be under estimate of actual number of success trajectories.

Table 10: Summary of trajectories sampled from SWE-Gym.

This gives us in in total 106 + 19 + 67 + 299 = 491 success trajectories, which forms our final training trajectories $D_2$.

### B.5. MoatlessTools ORM Prompt

The following is a pseudo-code that generates a prompt for MoatlessTools Verifier (ORM), which is modified from (Zhang et al., 2024a). Unlike (Zhang et al., 2024a), which relies on proprietary models like Claude-3.5-Sonnet for context extraction, we obtain context directly from the agent's trajectory being evaluated.

```
SYSTEM_MESSAGE = """You are an expert in python for software engineering and code
↪   review. Your responsibility is to review the patches generated by language
↪   models to fix some issues and provide feedback on the quality of their
↪   code."""

USER_MESSAGE="""I want you to evaluate an LLM-generated candidate patch that
↪   tries to resolve an issue in a codebase.

To assist you in this task, you are provided with the following information:
 - You are given an issue text on a github repository (wrapped with
 ↪   <issue_description></issue_description>).
 - You are also given some identified code spans that are relevant to the issue.
   Each code span is wrapped with <code_span file_path=FILE_PATH
    ↪   span_id=SPAN_ID></code_span> tags, where FILE_PATH is the path to the
    ↪   file containing the code span, and SPAN_ID is the unique identifier for
    ↪   the code span.
```

```
      Each code span also comes with the line numbers for you to better understand
      ↪   the context. It's possible that the code span are not sufficient to fix
      ↪   the issue, adjust your score accordingly.
   - You are given the candidate patch that tries to resolve the target issue.
      For your convenience, you are given the hunks of original code and the code
      ↪   after applying the patch.
      The code before the patch is wrapped with <before_patch></before_patch> and
      ↪   the code after the patch is wrapped with <after_patch></after_patch>.
      Note that the file names in before_patch starts with 'a/' and the file names
      ↪   in after_patch starts with 'b/'.

<issue_description>
{issue_text}
</issue_description>

<before_patch>
{before_patch}
</before_patch>

<after_patch>
{after_patch}
</after_patch>

{code_spans}

Response in "True" or "False" for whether the patch has resolved the issue."""
```

## B.6. OpenHands ORM Prompt

The following is a pseudo-code that generates a prompt for OpenHands Verifier (ORM).

```
SYSTEM_MESSAGE = '''You are an expert judge evaluating AI assistant interactions.
↪   Your task is to determine if the assistant successfully resolved the user's
↪   request.

Key evaluation criteria:
1. Did the assistant complete the main task requested by the user?
2. Did the assistant handle all edge cases and requirements specified?
3. Were there any errors or issues in the final solution?
4. Did the assistant verify the solution works as intended?

Respond only with "<judgement>YES</judgement>" or "<judgement>NO</judgement>".'''

USER_MESSAGE = '''Please evaluate the following interaction between an AI
↪   assistant and a user:

=== INTERACTION LOG ===
''' + traj_str + '''
=== END INTERACTION ===

Based on the above interaction, did the assistant successfully resolve the user's
↪   initial request? Respond with YES or NO.'''
```

```
messages = [
    {'role': 'system', 'content': SYSTEM_MESSAGE},
    {'role': 'user', 'content': USER_MESSAGE},
    {'role': 'assistant', 'content': '<judgement>' + ("YES" if resolved else
    ↪  "NO") + '</judgement>'}
]
```

The last assistant messages that contains judgement is only provided during training time. At inference time, the trained verifier is responsible predicting the probability of 'Yes' and 'No'.

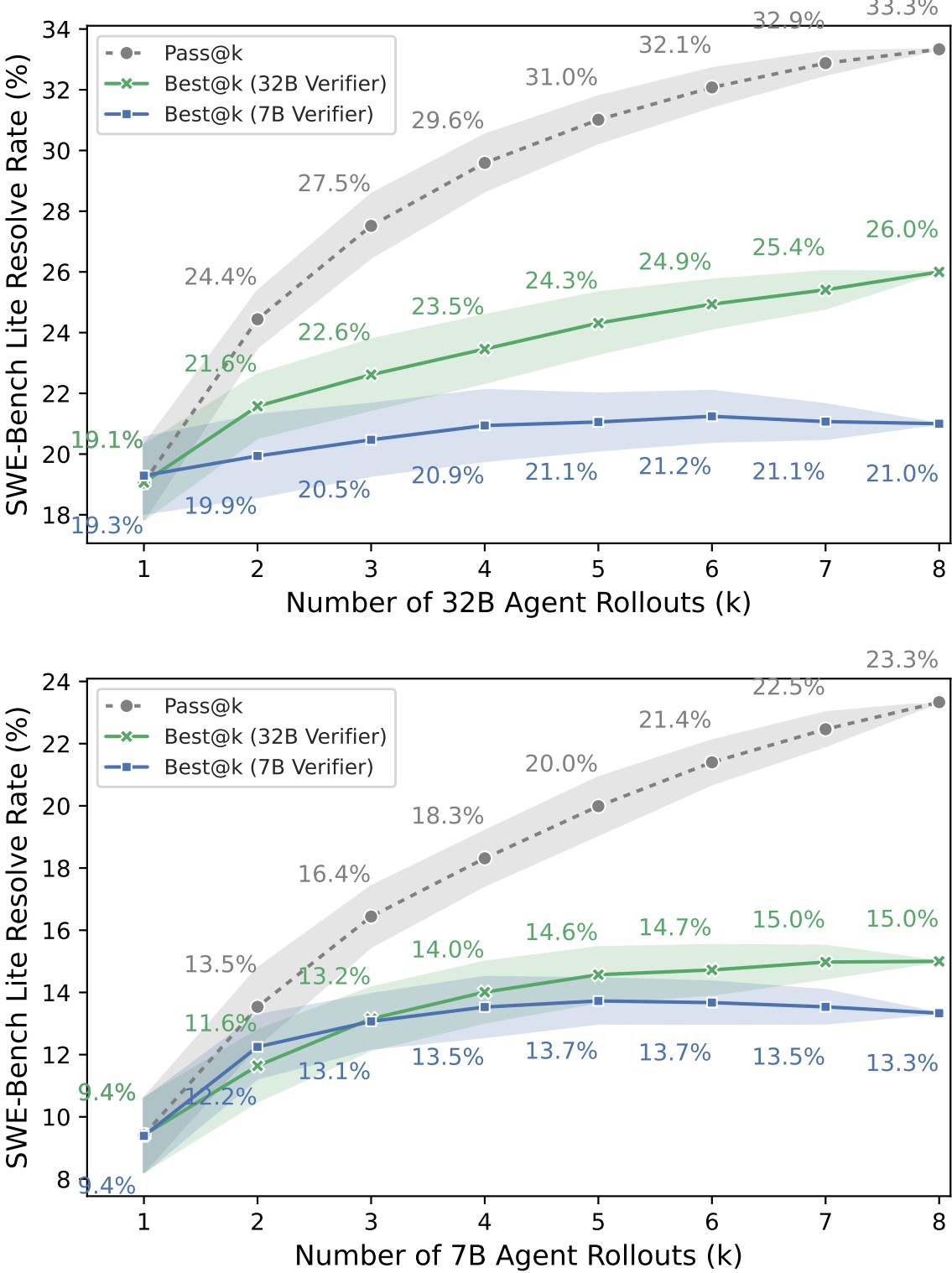

Figure 7: Scaling inference-time compute for MoatlessTools Agents (7B and 32B) with their corresponding learned verifiers. Temperature $t = 0.5$.

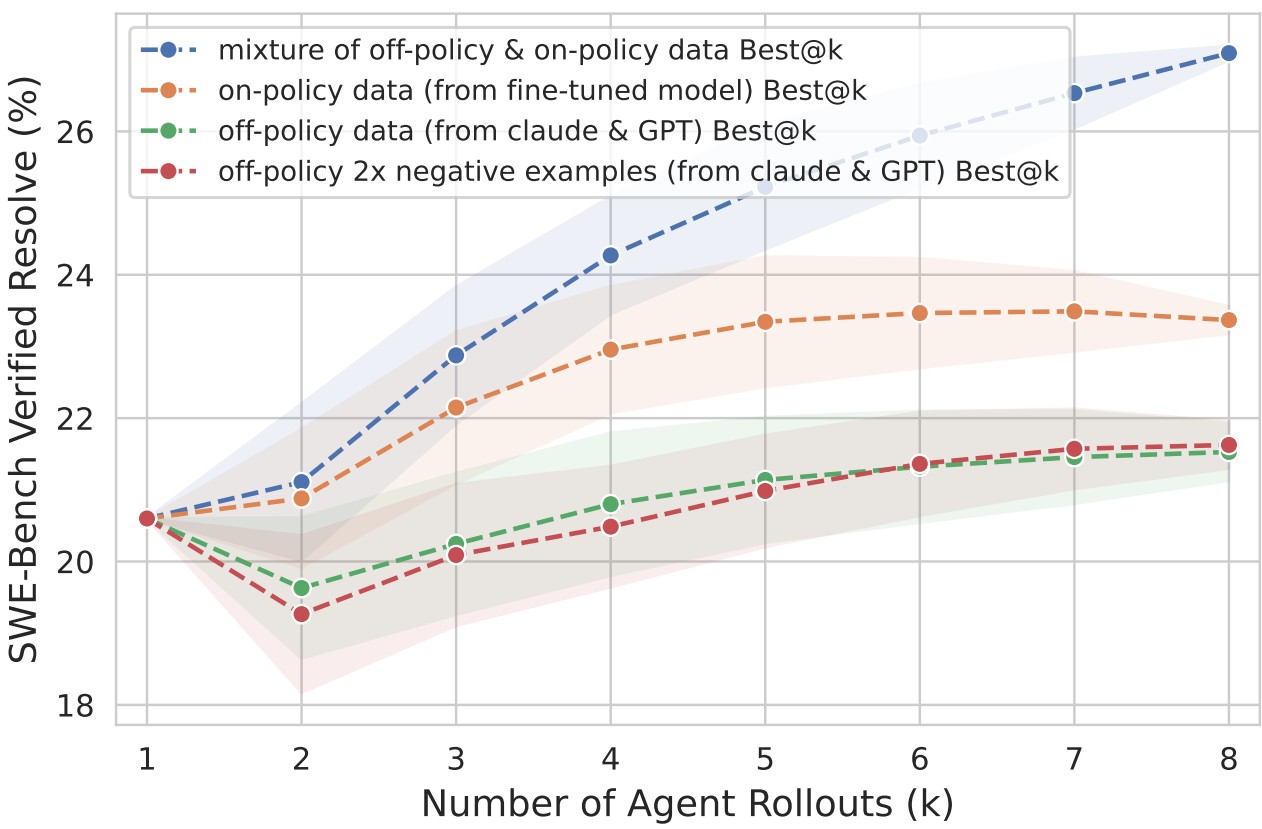

Figure 8: Ablation study for verifier training (§5.1.1). Performances are evaluated on SWE-Bench Verified. Both the agent and the verifier are `Qwen2.5-Coder-Instruct-32B` model fine-tuned on the corresponding dataset. OpenHands (Wang et al., 2024c) is used as the agent scaffold.

## Different strategies of scaling training trajectories

Figure 9: Comparison of three data sampling approaches using 32B LM: scaling trajectories (dedup.), scaling unique task instances, and scaling repositories (§5.2).

