# OpenReview forum: "Training Software Engineering Agents and Verifiers with SWE-Gym"
_ICML.cc/2025/Conference — ICML 2025 poster_

### Official Review · Reviewer_LJa6 · 2025-02-23

**Overall Recommendation:** 2

**Summary:**

The paper introduces SWE-Gym, a novel training environment specifically designed for developing software engineering agents. The environment comprises 2,438 real-world Python task instances extracted from GitHub issues; each instance includes a codebase, an executable runtime environment with pre-installed dependencies, and a set of unit tests for verification. The authors leverage SWE-Gym to train language model (LM) based agents and verifiers, demonstrating significant improvements in task resolution rates on standard benchmarks such as SWE-Bench Lite and SWE-Bench Verified. Key contributions include:

• A large-scale, realistic dataset that bridges the gap between existing synthetic or limited real-world benchmarks, enabling end-to-end training of agents that can handle complex repository-level software tasks.

• The development and evaluation of two agent scaffolds—one based on general-purpose prompting (OpenHands) and another on a specialized workflow (MoatlessTools). Using these, the authors fine-tune LM agents via rejection sampling fine-tuning, which leads to substantial improvements (up to 19% absolute gains in resolve rate).

• A novel approach to inference-time scaling where a verifier model is trained to estimate the success probability of candidate agent trajectories. By sampling multiple solutions and selecting the best one according to the verifier, they further boost performance to new state-of-the-art levels (achieving 32.0% and 26.0% on SWE-Bench Verified and Lite, respectively).

**Claims And Evidence:**

While many experimental claims, such as improved agent performance through fine-tuning and inference-time scaling, are well-supported, the submission’s claim that SWE-Gym is a significant contribution primarily due to its larger scale compared to SWE-Bench is not clearly justified. The paper does not provide sufficient evidence or ablation studies to demonstrate that a larger dataset—and the inclusion of executable environments—is necessary or offers unique benefits over existing datasets, leaving that claim problematic.

**Essential References Not Discussed:**

The paper do not discuss related works which also evaluated on swe-bench:

[1] Yang J, Jimenez C E, Wettig A, et al. Swe-agent: Agent-computer interfaces enable automated software engineering[C]//The Thirty-eighth Annual Conference on Neural Information Processing Systems. 2024.

[2] Xia C S, Deng Y, Dunn S, et al. Agentless: Demystifying llm-based software engineering agents[J]. arXiv preprint arXiv:2407.01489, 2024.

**Experimental Designs Or Analyses:**

The evaluation provides evidence that the training method is effective to some extent, yet there are experimental design concerns. Specifically, while the results suggest improvements using SWE-Gym, the study lacks direct comparisons with existing methods evaluated on SWE-Bench and does not explore the impact of training on subsets of SWE-Bench itself. This omission makes it difficult to isolate the unique contribution of SWE-Gym and assess whether similar gains could be achieved using parts of the established benchmark dataset.

**Methods And Evaluation Criteria:**

The choice of SWE-Gym over SWE-Bench as a primary training dataset is not convincingly justified. SWE-Bench already includes multiple verified versions and has undergone human validation, ensuring high-quality benchmarks for evaluating agent performance.

**Other Comments Or Suggestions:**

N/A

**Other Strengths And Weaknesses:**

N/A

**Questions For Authors:**

1. Could you clarify why a larger dataset (SWE-Gym) with executable environments is necessary, given that SWE-Bench already offers multiple, human-verified versions?
2. Have you conducted ablation studies or direct comparisons to evaluate the benefit of training on SWE-Gym versus using subsets of SWE-Bench?
3. Can you provide comparisons against existing methods evaluated on SWE-Bench to isolate the unique contributions of your training approach?
4. How do you justify the chosen evaluation criteria in light of the human verification already present in SWE-Bench, and what additional benefits does SWE-Gym provide?

**Relation To Broader Scientific Literature:**

The paper is realted SWE-bench, SWE-agent and so on.

**Theoretical Claims:**

The paper do not have theoretical claims.

---

> ### Author Rebuttal · Authors · 2025-04-01
>
> > Could you clarify why a larger dataset (SWE-Gym) with executable environments is necessary, given that SWE-Bench already offers multiple, human-verified versions?
>
> We’d like to clarify that SWE-Bench doesn’t include executable environments or unit tests for its training split. This makes it impossible to use for learning algorithms that train models through real-world action execution and observation. And training on the test split will introduce data
> contamination and invalidate our results.
>
> SWE-Gym addresses this fundamental gap by providing a separate, complementary dataset with fully executable environments that allows us to train agents on real-world software engineering tasks while maintaining the integrity of SWE-Bench as a clean evaluation benchmark. This separation is crucial for accurately measuring progress of software engineering agents.
>
> > Can you provide comparisons against existing methods evaluated on SWE-Bench to isolate the unique contributions of your training approach?
>
> Because our training approaches are focused on training models in real-world software engineering tasks, this requires execution environments and test cases for training task instances. SWE-Bench only includes environments and test cases for its test set, not its training set. Thus, we did not conduct these ablation studies as training on any subset of SWE-Bench’s test set would be problematic.
>
>
>
> > How do you justify the chosen evaluation criteria in light of the human verification already present in SWE-Bench, and what additional benefits does SWE-Gym provide?
>
> We want to clarify that throughout our paper, we exclusively use SWE-Bench (both Verified and Lite versions) as our evaluation framework. This choice is deliberate as SWE-Bench provides human-verified test cases that serve as a reliable, standardized benchmark for measuring agent performance. SWE-Gym complements this by uniquely providing an effective training dataset with executable environments that enable end-to-end agent training without contaminating our evaluation data.
>
>
>
> > Essential References Not Discussed and Comparisons against existing methods evaluated on SWE-Bench
> We appreciate the reviewer pointing out these references. We would like to clarify that we have discussed both references in our paper: SWE-agent [1] in line 102 and Agentless [2] in line 093.
>
> However, we acknowledge that our comparison with these works could be more comprehensive. In the next version of our paper, we will expand our analysis to include a more detailed comparison of our results with these frameworks. Essentially, although these approaches demonstrate that better prompts and agent scaffolds can enhance performance, our work shows that horizontally, end-to-end training—without relying on manual prompt design—can yield even greater improvements.
> We will also present their performance on SWE-Bench to better contextualize our contributions.
>
> Additionally, for concurrent works on SWE Agent training, we include a detailed comparison in appendix section A.

---

### Official Review · Reviewer_Be56 · 2025-03-10

**Overall Recommendation:** 4

**Summary:**

This paper introduces SWE-Gym, the environment for training software engineering (SWE) agents. SWE-Gym contains 2,438 real-world Python tasks from 11 popular GitHub repositories, each equipped with pre-installed dependencies, executable runtime environments, unit tests, and natural language task descriptions. The authors demonstrate SWE-Gym's effectiveness by using it to train language model-based agents through rejection sampling fine-tuning, achieving significant improvements in resolve rate on the SWE-Bench Verified and Lite test sets. The paper also explores inference-time scaling through verifiers trained on agent trajectories sampled from SWE-Gym, showing that when combined with fine-tuned SWE agents, they achieve state-of-the-art performance of 32.0% and 26.0% on SWE-Bench Verified and Lite respectively. The authors publicly release SWE-Gym, the trained models, and agent trajectories to facilitate further research.

**Claims And Evidence:**

The claims made in the paper are generally well-supported by evidence.

**Essential References Not Discussed:**

The paper covers most essential references in the field.

**Experimental Designs Or Analyses:**

- Agent training experiments: The authors clearly specify model sizes, training procedures, and hyperparameters. The comparison between different agent scaffolds (OpenHands vs. MoatlessTools) provides valuable insights into the effectiveness of SWE-Gym across different agent architectures.

- Verifier experiments: The authors explore different training data compositions for verifiers, showing how mixing on-policy and off-policy trajectories affects performance.

- Scaling experiments: The three different scaling approaches (trajectory, instance, and repository scaling) are well-designed to isolate the impact of different aspects of training data.

- Statistical rigor: The authors report standard deviations for key metrics (Table 3), enabling assessment of result reliability.

- **One minor concern** is that the computational budget constraints limited the number of training trajectories to 491, which may affect the generalizability of some findings.

**Methods And Evaluation Criteria:**

The methods and evaluation criteria are appropriate and well-designed for the problem:

- Dataset construction: The authors detail a rigorous process for creating SWE-Gym, including repository selection criteria, versioning, and environment setup. The validation of instances using execution-based verification ensures high-quality training data.

- Evaluation metrics: The use of standard SWE-Bench metrics (resolve rate, empty patch rate) provides consistency with prior work. Additional metrics like "stuck in loop" percentage offer valuable insights into agent behavior improvements.

**Other Comments Or Suggestions:**

Consider expanding the discussion on how SWE-Gym could be extended to other programming languages beyond Python in future work.

**Other Strengths And Weaknesses:**

Strengths:
- Practical contribution: SWE-Gym addresses a critical need in the field by providing a standardized, reproducible environment for training SWE agents.

- Comprehensive experimentation: The paper explores multiple dimensions (model size, agent scaffold, training data composition).

- Scaling analysis: The clear demonstration of scaling behaviors with both training data and inference compute.

Weaknesses:

- Limited exploration of alternative fine-tuning methods: While rejection sampling is effective, comparing with other approaches like PPO or DPO would strengthen the paper.

- Computational constraints: The limited number of training trajectories (491) may not fully reveal the potential of SWE-Gym, though the authors acknowledge this limitation.

- Task diversity analysis: While the paper mentions task distribution across repositories (Fig. 2), deeper analysis of how different task types benefit from training could provide additional insights.

**Questions For Authors:**

See "Other Strengths And Weaknesses".

**Relation To Broader Scientific Literature:**

The authors appropriately cite relevant prior work and clearly articulate how SWE-Gym addresses a critical gap in the literature.
- SWE agent development: The authors contextualize their work within recent advances in SWE agents, highlighting the limitations of current approaches due to lack of suitable training environments.

- Agent scaffolds: The paper discusses different agent design philosophies (specialized workflows vs. general-purpose prompting) and evaluates SWE-Gym's effectiveness across both paradigms.

- Post-training methods: The authors connect their work to broader trends in LLM fine-tuning techniques, including trajectory filtering approaches.

- Verifier models: The paper builds on outcome-supervised reward modeling and applies it to the software engineering domain.

**Theoretical Claims:**

The paper does not make formal theoretical claims requiring proof verification. The claims are empirical in nature, focusing on the effectiveness of SWE-Gym for training agents and verifiers.

---

> ### Author Rebuttal · Authors · 2025-04-01
>
> >  computational budget constraints limited the number of training trajectories to 491, which may affect the generalizability of some findings.
>
> We would like to emphasize that our results represent the state-of-the-art open-model results at the time of submission, and used a substantial compute budget exceeding $30K. The challenge of collecting more training trajectories stems from the inherent computational intensity of agent training research, particularly when working with real-world software tasks that require full execution environments.
>
> Importantly, our scaling experiments in Sections 5.1 and 5.2 demonstrate consistent log-linear scaling behavior, which provides strong evidence for the generalizability of our findings beyond the current dataset size. Our experiments in Section 5.2 suggest that the task diversity of SWE-Gym is not a limiting factor in further model improvements.We believe these results offer valuable insights that will motivate and guide future open research in this direction.
>
> Additionally, we note that these 491 trajectories are positive ones used for agent training. In fact, we used over 1000 trajectories (both positive and negative) for training the verifiers.
>
> > Limited exploration of alternative fine-tuning methods
>
> We appreciate the reviewer's insightful suggestion regarding alternative fine-tuning methods. We would like to clarify that the primary contribution of our work is the SWE-Gym dataset itself, which addresses a critical gap in the field by providing executable environments for real-world software engineering tasks. While we demonstrate the dataset's effectiveness through recently-proposed approaches for model improvement via agent training and test-time scaling, these implementations serve as solid baselines rather than exhaustive explorations of optimal training techniques.
>
> As one of the first papers to study SWE agent training on real-world tasks, we deliberately established strong baseline models using well-understood training approaches (Zelikman 2022, Singh 2023, Pan 2024) to provide clear evidence of SWE-Gym's effectiveness. Our results show substantial performance improvements (up to 19% absolute gains in resolve rate) using these straightforward methods, which we believe validates SWE-Gym's value as a training resource.
>
> We agree that exploring more sophisticated training paradigms represents a promising direction for future research. By releasing our dataset, trained models, and agent trajectories publicly, we aim to facilitate such explorations by the broader research community.
>
> > How SWE-Gym could be extended to other programming languages beyond Python in future work.
>
> We thank the reviewer for this valuable suggestion. In our future work section, we will add a comprehensive discussion on ways to extend SWE-Gym beyond Python. We envision two promising approaches: (1) establishing a collaborative community effort to systematically collect and curate datasets across multiple programming languages; or (2) developing specialized environment-setup language model agents that can automatically analyze repositories, identify dependencies, and construct executable environments for diverse programming languages.
>
> > I see two addition as compared to boarded literature i) Addition of verifiers in fine-tuning of agent and ii) adding unit tests for part of training data
>
> We thank the reviewer for pointing out these contributions. We will update the related works section to include a more detailed comparison of our results with these literature.

---

> > ### Comment · Reviewer_Be56 · 2025-04-02
> >
> > Thank you for the author's reply. The author addressed most of my concerns. I read the paper again carefully and I think the community needs this dataset to handle real-world agent tasks (although the dataset is a bit small). Therefore, I raised my score to **4**.

---

### Official Review · Reviewer_aRo7 · 2025-03-20

**Overall Recommendation:** 1

**Summary:**

The paper proposes SWE-Gym, which is a training environment for coding agents tasked to resolve GitHub issues.
They provide a collection of 2438 python-based SWE tasks.
They used filtered fine-tuning and showed improvement by fine-tuning LLMs in their training environment.
Finally, to show effectiveness authors compared resolved rates on swebench verified and lite benchmarks.
They also trained verifiers which will be also beneficial for training via RL.

**Claims And Evidence:**

Yeah, claims are generally clear and backed with experiments.

**Essential References Not Discussed:**

None

**Experimental Designs Or Analyses:**

Yes.
Training with SWE-gym and Scaling agent performance, both seems reasonable.

**Methods And Evaluation Criteria:**

They use Resolve Rate (%), Empty Patch Rate (%), stuck in the loop rate, pass@k and best@k as criteria for improvement which makes sense. I don't think pass@k with k>=2 is of much use though.

However, to better demonstrate the effectiveness, authors can also add precision and recall which can demonstrate the effectiveness of intermediate steps as well. This will also be convincing to demonstrate effectiveness due to the verifier's rewards.

**Other Comments Or Suggestions:**

In the abstract, can you add the model name and the number of parameters along with improvement numbers?

**Other Strengths And Weaknesses:**

Strengths:
- The paper is clearly written and easy to parse
- Enhancing agent's performance via verifier.
- Scaling experiments

Weakness:
- I think Novelty is limited given that swe-bench training data already provide training data (if unit tests are created for swe-bench, then I don't see what's the difference)
- Doesn't include precision and recall which are essential to understand the trajectories.
- Missing analysis on how untrained LLMs will perform on the training data? Is this training data verified or not, i.e., does it contain sufficient information for the issues to be resolved? It may be the case that LLM learns to hallucinate if trained on this since there may be training data which doesn't have sufficient information
- only ~2.5K training samples which I think are small for RL-type training

**Questions For Authors:**

In Table 3, how many runs are used for showing the confidence intervals?

Would be good if confidence intervals could be added to other results or comment on them; given the stochasticity of LLMs, its difficult to conclude from a single-digit number.

###Update after rebuttal:

All the weaknesses are still intact and the author's response does not address them constructively.
- Especially, they agree that the difference to the SWE-bench is limited as was pointed out in the weakness initially that if unit tests are included in the swe-bench then there is no difference.
- I think the response to stochasticity (or confidence interval) should be "evidence with data" as compared to saying "We believe the gap is already enough". I would appreciate if the authors were scientifically rigorous.
- Also, authors agree that they need significant work, especially in coming up with metrics like precision/recall in their data set before it is useful (or better than existing benchmarks).

In response to this, I suggest rejecting the paper. It's also beneficial for authors to submit a complete and usable work.

**Relation To Broader Scientific Literature:**

I see two addition as compared to boarded literature i) Addition of verifiers in fine-tuning of agent and ii) adding unit tests for part of training data

**Theoretical Claims:**

Experimental paper; No theoretical claims.

---

> ### Author Rebuttal · Authors · 2025-04-01
>
> We thank Reviewer aRo7 for their insightful feedback. Below, we address the primary concerns and suggestions provided.
>
> > I think Novelty is limited given that swe-bench training data already provide training data
>
> To clarify, SWE-Bench doesn’t include unit tests or executable environments for their training data. Its training data is insufficient for effective agent training because it lacks both executable environments and test case for those instances. Thus, we put in the work to create a suitable training environment. We decided to focus on our own set of repositories, rather than the ones included in SWE-bench, to avoid contamination concerns.
>
> Our work is one of the first to study *training* SWE agents on real-world software engineering tasks, which is a significant departure from existing work that focuses on evaluation or prompting-based agents. We also achieve state-of-the-art open-model results on SWE-Bench, with log-linear training and test-time scaling results.
>
> These distinctions make SWE-Gym a novel and valuable contribution to the field of software engineering agents.
>
> > Precision and Recall Analysis:
>
> We appreciate the suggestion to include precision and recall metrics. While our evaluation follows established protocols from prior work (Cobbe et al., 2021), we agree that precision-recall curves would provide more comprehensive insights into our verifiers' performance. We are currently working on these curves and will soon follow up with the updated results.
>
> Regarding intermediate step evaluation, developing effective process rewards for software engineering agents is still an open problem. We would welcome the reviewer's insights on potential approaches to this problem, and would love to incorporate these in the future work.
>
> > Clarification Regarding pass@k:
>
> We clarify that pass@k (k≥2) solely serves as a reference point to compare our learned verifiers against an oracle verifier that always selects the optimal solution. This metric was not used for our primary results or for comparisons with other methods.
>
> > Is this training data verified or not, i.e., does it contain sufficient information for the issues to be resolved? It may be the case that LLM learns to hallucinate if trained on this since there may be training data which doesn't have sufficient information
>
> Our data is as verified as in the original SWE-Bench paper. As described in Lines 152-186, following SWE-Bench’s task construction pipeline, we only keep around issues that have a gold-standard PR, which indicates that the issue was able to be resolved by a human programmer based on the information provided. Also, as shown in Table 9 in Appendix, claude 3.5 sonnet without any SWE-Gym specific training achieves the reasonable performance of 29.1% on SWE-Gym Lite within 50 turns, suggesting that a significant proportion of the task in SWE-Gym is solvable.
> Furthermore, our ultimate evaluation of our trained model is on SWE-Bench, not SWE-Gym, which is quite out of distribution from SWE-Gym, so any repo-specific hallucination learned through SWE-Gym wouldn't explain our improved SWE-Bench performance.
> We’d be happy to add follow-up experiments if the reviewer has any thoughts on validating this hypothesis.
>
> > In Table 3, how many runs are used for showing the confidence intervals? Would be good if confidence intervals could be added to other results or comment on them; given the stochasticity of LLMs, its difficult to conclude from a single-digit number.
>
> To mitigate the stochasticity of LLMs, we apply a consistent random seed and sampling temperature of 0 across all experiments in Table 3. We use a bootstrap test to estimate the standard deviation by sampling 1000 different subsets of the evaluation result.  Regardless of confidence estimation, we believe the performance gap before/after fine-tuning is significant.
> For experiments with higher stochasticity, we plot error regions in Figures 3 and 4 to represent confidence intervals, as described in lines 357-365. Following Lightman et al. (2023), we estimate the uncertainty as detailed in Appendix B.2.
>
> > In the abstract, can you add the model name and the number of parameters along with improvement numbers?
>
> We thank the reviewer for the suggestion. We will update the abstract to include the model name and the number of parameters.
>
> > only~2.5K training samples
>
> We kindly request the reviewer to refer to our first response to Reviewer BE56, where we discuss how our dataset is already effective for model training and achieves state-of-the-art results. Importantly, our experiments in Section 5.2 directly show that the task diversity of SWE-Gym is not a limiting factor in further model improvements.

---

### Decision · Program_Chairs · 2025-05-01

**Decision:**

Accept (poster)

**Comment:**

This paper introduces SWE-Gym, a novel environment featuring executable setups and unit tests for training software engineering agents on real-world Python tasks. The authors demonstrate its utility by fine-tuning language model agents and training verifiers, achieving state-of-the-art results for open-weight models on the established SWE-Bench benchmark. Strengths highlighted by reviewers include the practical contribution of a much-needed training resource (Be56), comprehensive experiments with scaling analysis (Be56), and clear writing (aRo7). Key concerns revolved around the perceived novelty and necessity compared to SWE-Bench (aRo7, LJa6), the dataset size (aRo7, Be56), and the handling of evaluation metrics and stochasticity (aRo7).

Addressing the primary concerns, the authors effectively clarified that the SWE-Bench training split lacks the executable environments and tests required for agent training via environment interaction, making SWE-Gym a necessary contribution distinct from the SWE-Bench evaluation set. Using the SWE-Bench test set for training would constitute data contamination. While the dataset size and number of training trajectories collected are limited by significant computational costs, the authors demonstrate strong empirical results, including state-of-the-art performance and positive scaling trends, validating the environment's effectiveness with the current data (Be56). Concerns about specific metrics like precision/recall (aRo7) or alternative fine-tuning methods (Be56) point towards valuable future work but do not invalidate the current contributions evaluated using standard metrics.

Overall, SWE-Gym represents a significant step forward by providing the first large-scale, executable environment for training agents on realistic software engineering tasks, complementing existing evaluation benchmarks like SWE-Bench. The demonstrated state-of-the-art results achieved through training within SWE-Gym underscore its value. Despite limitations inherent in costly agent-environment interaction research, such as dataset scale, the work presents a solid contribution, releasing valuable resources (environment, models, trajectories) to the community. The critiques regarding novelty relative to SWE-Bench appear based on a misunderstanding of benchmark limitations, and other concerns represent avenues for future exploration rather than fundamental flaws. The paper meets the bar for acceptance at ICML.